# Oxygen Gradient Induced in Microfluidic Chips Can Be Used as a Model for Liver Zonation

**DOI:** 10.3390/cells11233734

**Published:** 2022-11-23

**Authors:** Shahrouz Ghafoory, Christina Stengl, Stefan Kopany, Mert Mayadag, Nils Mechtel, Brennah Murphy, Sebastian Schattschneider, Niklas Wilhelmi, Stefan Wölfl

**Affiliations:** 1Institute for Pharmacy and Molecular Biotechnology, Im Neuenheimer Feld 364, 69120 Heidelberg, Germany; 2Medical Physics in Radiation Oncology, German Cancer Research Center (DKFZ), Im Neuenheimer Feld 280, 69120 Heidelberg, Germany; 3Faculty of Medicine, University of Heidelberg, Im Neuenheimer Feld 672, 69120 Heidelberg, Germany; 4Heidelberg Institute for Radiation Oncology (HIRO), National Center for Radiation Research in Oncology, Im Neuenheimer Feld 280, 69120 Heidelberg, Germany; 5The Wistar Institute, Philadelphia, PA 19104, USA; 6Microfluidic ChipShop, GmbH Stockholmer Str. 20, 07747 Jena, Germany

**Keywords:** microfluidic chips, oxygen, albumin, Hif1α, hypoxic condition, liver, HepG2, in vitro, CoCl_2_, DFX

## Abstract

Availability of oxygen plays an important role in tissue organization and cell-type specific metabolism. It is, however, difficult to analyze hypoxia-related adaptations in vitro because of inherent limitations of experimental model systems. In this study, we establish a microfluidic tissue culture protocol to generate hypoxic gradients in vitro, mimicking the conditions found in the liver acinus. To accomplish this, four microfluidic chips, each containing two chambers, were serially connected to obtain eight interconnected chambers. HepG2 hepatocytes were uniformly seeded in each chamber and cultivated under a constant media flow of 50 µL/h for 72 h. HepG2 oxygen consumption under flowing media conditions established a normoxia to hypoxia gradient within the chambers, which was confirmed by oxygen sensors located at the inlet and outlet of the connected microfluidic chips. Expression of Hif1α mRNA and protein was used to indicate hypoxic conditions in the cells and albumin mRNA and protein expression served as a marker for liver acinus-like zonation. Oxygen measurements performed over 72 h showed a change from 17.5% to 15.9% of atmospheric oxygen, which corresponded with a 9.2% oxygen reduction in the medium between chamber1 (inlet) and 8 (outlet) in the connected microfluidic chips after 72 h. Analysis of Hif1α expression and nuclear translocation in HepG2 cells additionally confirmed the hypoxic gradient from chamber1 to chamber8. Moreover, albumin mRNA and protein levels were significantly reduced from chamber1 to chamber8, indicating liver acinus zonation along the oxygen gradient. Taken together, microfluidic cultivation in interconnected chambers provides a new model for analyzing cells in a normoxic to hypoxic gradient in vitro. By using a well-characterized cancer cell line as a homogenous hepatocyte population, we also demonstrate that an approximate 10% reduction in oxygen triggers translocation of Hif1α to the nucleus and reduces albumin production.

## 1. Introduction

A constant supply of oxygen is crucial for multicellular life on earth. Interestingly, while oxygen must be supplied in a narrow range of concentrations to prevent the damaging effects of hypoxia or hyperoxia [1], cells can also adjust to different oxygen concentrations, and even exploit differences in oxygen supply for optimized metabolic functions. The proteins of the hypoxia-inducible factor (Hif) family are key regulators to maintain oxygen homeostasis at both systemic and cellular levels. These proteins are conserved during evolution in all species, from worms and flies to vertebrates [2]. Hif1 is a heterodimer consisting of one subunit of Hif1α, Hif2α, or Hif3α, and the β-subunit—a constitutive nuclear protein [3]. The Hif1α subunit is the most extensively studied isoform and plays a key regulatory role in Hif1 function. Under normoxic conditions, the Hif1α subunit is continuously synthesized and degraded by the ubiquitin-proteasome system in the cytoplasm. However, under hypoxic conditions, the Hif1α subunit is protected against ubiquitination and degradation, and subsequently translocated to the nucleus [4,5]. In the nucleus, the Hif1α and β subunits dimerize and bind to several hypoxia-responsive elements (HRE) in the 3′ enhancer regions of up to 70 genes and activate their expression [1,6,7]. A wide range of activity has been reported in both normal and malignant cells under hypoxic conditions. Hif1 not only allows normal and malignant cells to survive in low oxygen level areas by shifting glucose metabolism from oxidative phosphorylation to anaerobic glycolysis, but also leads to enhancing angiogenesis to increase oxygen access. Hif1α may also influence the production of the plasma protein albumin in hepatocytes [8,9,10,11,12]. Albumin is an essential component for maintaining oncotic pressure and transporting hydrophobic endogenous and exogenous substances, such as hormones, in the blood [13]. In liver hepatocytes, albumin is produced to different extents depending on the location in the liver acinus. Periportal hepatocytes express the highest levels of albumin and this expression decreases steadily towards the pericentral area. Thus, albumin can be used as an indicator of metabolic zonation [14]. Hepatocytes themselves account for about 60% of the complex liver architecture, which are arranged hexagonally in so-called lobules. Blood entering the liver through the hepatic artery or portal vein flows in one direction to the central vein. This unidirectional blood flow coupled with hepatocyte oxygen consumption creates an oxygen gradient in the liver microenvironment with increasing hypoxic conditions from the periportal to the pericentral region [15,16]. Conventionally, deferoxamine (DFX), an iron chelator, and cobalt chloride (CoCl_2_) are used to induce hypoxia in cell culture systems, as these agents have been shown to induce Hif1α activity with the same signaling pathway as hypoxic conditions [17,18]. While useful, this approach fails to recapitulate a hypoxia gradient in vitro.

The concept of positioning and culturing cells inside a microfluidic system started with the creation of microprinted and injection-molded microfluidic chips (MCs) [19], which were soon combined with on-line sensing technology and first used to simulate systemic interactions between organ-mimicking compartments [20,21,22]. Given the potential to supplement animal studies and improve predictions of clinical outcomes, the field of organs-on-a-chip has been continuously advancing [23], with various designs depending on the questions addressed [24]. Thus, MCs have gained increasing importance in in vitro modeling of metabolic activities inside and between organs. We hypothesized that the basic physiology of an acinus could be modeled in a chip system with a unidirectional medium flow and applied for the simulation of hepatic zonation. The model should enable us to investigate the cellular mechanisms underpinning the spatially driven differential metabolic phenotypes within one single cell type as found in the acinus, while at the same time providing sufficiently large numbers of cells for each condition to enable further analysis of the different phenotypes.

In this study, we employed four connected MCs, with two chambers each, and passed media unidirectionally through the chambers, thereby modeling the spatial characteristics of oxygen supply along liver sinusoids. With this system, we were able to reproduce the oxygen gradients observed in the liver acini and test changes in gene expression in a well-characterized hepatoblastoma-derived cell line along an oxygen gradient. As an additional control, we also used DFX, an iron chelator, and CoCl_2_ to induce Hif1α activity and independently mimic hypoxic conditions in the system.

## 2. Materials and Methods

### 2.1. Microfluidic Chips

A two-chamber interconnected MC, consisting of three Mini Luer inlets and three Mini Luer outlet ports, was used during this study (product Code: 10,001,062, Lot: HH1 19,292; microfluidic ChipShop GmbH, Jena, Germany). It is made of polystyrene with a thickness of 1.5 mm and a cover foil (125.5 µm) overall 1.625 mm. The area of the whole chip is 1925.25 mm^2^, the area of one chamber is 189.5 mm^2^, and the volume of one MC is 106.2 μL. In each MC, two outlets and two inlets were blocked with Mini Luer single plug-Fluidic 334 green (product code 10,000,052; microfluidic ChipShop GmbH, Jena, Germany) and HepG2 cells were seeded via the remaining inlet or outlet at 7 × 10^6^ cells/mL in each MC separately. These single MCs were incubated in a standard tissue culture incubator at 37 °C, 5% CO_2_, and 95% humidity overnight to allow the cells to bind to the bottom of the chambers [25]. Later, one to four MCs with eight chambers in total were connected in a series using tubes made of silicone with the material code Silastic^®^ Q7-4750 (Dow Corning). Two different types of tubing were used: (i) to connect MCs with pumps and waste (inlet and outlet) we used (product code 10,000,031 with inside diameter (ID) 0.76 mm and outside diameter (OD) 1.65 mm, microfluidic ChipShop GmbH, Jena, Germany); (ii) for interconnecting sensor chip and MCs we used (product code 10,000,033 with ID 0.5 mm and OD 2.5 mm, microfluidic ChipShop GmbH, Jena, Germany) each connecting tube 3 cm long; together with red Mini Luer connectors-Fluidic 331 (product code 10000064, microfluidic ChipShop GmbH, Jena, Germany) (Figure 1A and Appendix A). Moreover, all joints between the MCs, adapters, and tubes were sealed with TFC silicon kautschuk type 6 (Troll Factory Rainer Habekost e.K., Riede, Germany).

### 2.2. Oxygen Measurement

Oxygen sensors including OXY-1 ST (Fiber-optic trace oxygen meter based on dynamic quenching), which were connected to an O2 SensorPlug, were placed in an individually manufactured MC (product code 10001496, Lot: IC122432; microfluidic ChipShop GmbH, Jena, Germany) via an ST optical fiber (PreSens GmbH, Regensburg, Germany) and installed before and after MC connections in a sterile condition as shown in Figure 1B and Appendix A. Four different setups were tested with 1, 2, 3, and 4 MCs connected in a row. A low-pressure Nemesys syringe pump (Cetoni GmbH, Korbußen, Germany) in combination with a 5.0 mL syringe H 1/4″-28 m Tubing Connector (ILS GmbH, Germany) were prepared to achieve a constant media flow rate of 50 µL/h in the connected chambers for 72 h in a standard tissue culture incubator. The data were collected by sensors every 5 min and processed by Presens measurement Studio (PMS2) software (PreSens GmbH, Regensburg, Germany) (Figure 1B and Appendix A). The final graph was drawn using GraphPad Prism 8.0.2.

### 2.3. Chemical Induction of Hypoxia

To confirm hypoxic conditions induced by the connected MCs, CoCl_2_, and DFX were used to induce a chemical hypoxic condition in one MC (Figure 1C). Either 100 µM CoCl_2_ or 250 µM DFX was dissolved in the media. Cells were seeded as previously explained and 24 h later, the media containing chemical agents was run through the MC at a rate of 50 µL/h for 72 h.

### 2.4. Measuring Cell Byproducts by Collecting Media Flowthrough

The normal media flowthrough was collected for the four different setups (MCs 1–4) at different time points (24 h, 48 h, and 72 h), saved in 1.5 mL Eppendorf tubes, and stored at −20 °C for later ammonia and glucose analysis. For both assays, samples were diluted 1:10 with distilled water. For the glucose assay, 10 µL of prepared standards and diluted samples were added to a flat bottom 96-well plate. A total of 240 μL of GOx mix (Tris 250 mM pH 8.0, GOx 50 mg/L, HRP 40 mg/L and o-dianisidine 100 mg/L) was added to each sample or standard and incubated for 10 min in the dark at room temperature (RT). The absorbance was measured at 450 nm by using the TECAN Sunrise microplate reader. An ammonia assay kit (Catalog number: AA0100-1KIT; Merck, Germany) and a 48-well plate were used to determine the ammonia levels as described by the manufacturer. Briefly, the samples were mixed with ammonia assay reagents and absorbance was measured at 340 nm by TECAN Sunrise microplate reader.

### 2.5. Cell Fixation

MCs were disconnected from the pump after 72 h and cells were fixed immediately using 500 µL 4% PFA per MC for 15 min, washed three times with PBS (500 µL per chip, 5 min at RT), and later stained with either immunofluorescent or fluorescent in situ hybridization methods.

### 2.6. IF Staining

Then, 500 µL of blocking buffer (BSA 3%, Triton™ X-100 0.8%, goat serum 3%) was introduced to the fixed cells in each MC for 1 h at RT followed by overnight incubation at 4 °C with 250 µL of primary binding buffer (BSA 1%, Triton™ X-100 0.5%, goat serum 1%) [26] with anti-Hif1α and albumin antibodies (Table 1). Later, the cells were washed with PBS three times for 5 min at RT, 250 µL of secondary fluorescence antibodies in PBS (Table 1) were added to each of the MCs and they were incubated for 2 h in the dark at RT. Later, the cells were washed with PBS in the dark at RT and images were taken with a fluorescence microscope (BZ-9000 BioRevo; Keyence, Osaka, Japan).

### 2.7. Antisense RNA Preparation

Total RNA was extracted from HepG2 cells (DNA, RNA, and protein purification kit; Macherey-Nagel, Dueren, Germany) followed by cDNA synthesis. Primers were designed by using a web-based tool and software designed by Nils Mechtel (https://nilsmechtel.shinyapps.io/primer_design/ (accessed on 15 April 2021)) (Table 2) based on the method introduced by Ghafoory et al. [27]. Target DNA sequences were amplified (normal PCR) and purified with NucleoSpin Gel and PCR clean-up kit (Macherey-Nagel, Dueren, Germany). Human HIF1A and ALBUMIN antisense RNAs were synthesized and labeled with Alexa Fluor 594 and Alexa Fluor 488 (FISH Tag™ RNA Green and Red Kits; Invitrogen by Thermo Fisher Scientific, Waltham, MA, USA), respectively, for fluorescence in situ hybridization (FISH).

### 2.8. FISH Staining

Then, 250 µL proteinase K (0.02 µg/µL) was introduced to fixed cells in each MC and incubated at 37 °C for 8 min followed by a 6 min incubation with 0.2% glycine at RT. Cells were postfixed with 4% PFA and 0.2% glutaraldehyde for 20 min at RT. Next, they were washed with 250 µL of PBS three times for 5 min. A total of 250 µL of hybridization mix (prewarmed at 95 °C for 5 min and cooled on ice immediately before use) [27] was added to each MC and the remaining unblocked channels were blocked with single plug–fluidic 334. All MCs were placed in a humid chamber and incubated at 69 °C for one hour. Later, the MCs were unblocked and 250 µL of ALBUMIN and HIF1A antisense probes (2 ng/µL) dissolved in a hybridization mix (prewarmed at 95 °C for 5 min and cooled on ice immediately before use) was added to each of them. MC channels were blocked again and incubated at 69 °C overnight in a humid chamber. The day after, MCs were shortly washed with 2× saline-sodium citrate buffer (SSC) for 5 min at RT and incubated with 50% formamide ⁄ 2× SSC at 65 °C for half an hour in the dark. Then, they were washed with PBS containing 0.1% Tween 3× for 5 min at RT in the dark.

### 2.9. Imaging

Images of all eight chambers were taken from at least three independent experiments with the same fluorescence exposure time and magnification for each channel on a Keyence BIOREVO microscope (BZ9000; Keyence, Osaka, Japan), a camera-based integrated fluorescence microscope using 20×, and 40× magnifications (Nikon objectives: Plan Apo ×20 NA 0.75 WD 1.00 mm, and Plan Apo ×40 NA 0.95 WD 0.14 mm). Keyence BIOREVO software was used for image processing and captured images were analyzed and fluorescence was quantified using ImageJ (version 1.53c).

### 2.10. Image Analysis

For the analysis of the translocation of Hif1α into the nucleus, background corrected, 8-bit images were used. First, the nuclei of the cells were identified using DAPI staining. Therefore, DAPI images were transformed into a binary mask of nuclei by a threshold range from 28 to 255. Next, this mask was overlaid with Hif1α stained cells of the same spot. The mean fluorescent intensity of each nucleus was then calculated in the Hif1α images. The cytoplasmic intensity was gained by measuring image intensity without nuclei intensity. Intensity values of both, cytoplasm and nucleus, were normalized to total fluorescence intensity. The fluorescence intensity of albumin was analyzed by retrieving the total intensity of 8-bit images stained with albumin. Next, the intensity was normalized to the DAPI-stained nuclei and the mean intensity of three independently measured images of chamber1 was set to 100% fluorescence intensity. ALBUMIN mRNA was analyzed similarly. For HIF1A mRNA total intensity of 8-bit images was measured and was also normalized to DAPI-stained nuclei. The intensity of chamber8 was set to 100%. All data are shown as mean ± standard error of the mean (SEM).

## 3. Results

### 3.1. Oxygen Level

HepG2 cells were seeded in one, two, three, or four connected MCs using tubes and fitting adapters, and oxygen levels were measured. The oxygen percentage of the inlet was measured to be 20.25% ± 0.06% at 30 min, decreased to 17.56% ± 0.09% after 48 h and stabilized at 17.55% ± 0.05% at 72 h (Table 3). Across the four connected MCs, a continuous decrease in oxygen level after 72 h was found. Looking at only one MC (chamber1) the oxygen level decreased from 20.09% ± 0.29% at 30 min to 17.31% ± 0.12% at 72 h. For all four connected chambers, the oxygen percentage decreased from 18.52% ± 1.26% measured after 30 min to 16.31% ± 0.23% at 12 h and stabilized at a steady state around 15.95% ± 0.36% at 72 h. These measurements indicate that in the four serially connected MCs, with a media pumping rate of 50 µL/h, oxygen was reduced in the media after 72 h by 9.2% between chamber1 and 8 (Figure 2).

### 3.2. Cell Byproducts in Media Flowthrough

Normal media flowthroughs from all four MCs were collected at different time points (24, 48, and 72 h) and analyzed for glucose and ammonia levels. The fresh media glucose concentration (4.5 g/L) was reduced within 72 h in all tested samples. For MCs 1, 2, 3, and 4, glucose concentration decreased to 3.85 ± 0.09 g/L, 3.54 ± 0.11 g/L, 2.61 ± 0.13 g/L, and 2.25 ± 0.04 g/L, respectively (Appendix A). Ammonia levels after 24 h for MCs 1, 2, 3, and 4 were 139,71 ± 56.28 µg/mL; 98.95 ± 13.49 µg/mL; 93.47 ± 33.65 µg/mL; and 91.34 ± 29.8 µg/mL, respectively, indicating a decrease in ammonia concentration from MC 1–4 (Appendix A). However, after 72 h, the ammonia concentration stabilized at 142.04 ± 44.74 µg/mL, 131.18 ± 36.33 µg/mL, 137.84 ± 48.04 µg/mL, and 127.68 ± 56.02 µg/mL, in MCs 1–4, respectively, and no significant difference was found in the ammonia concentration between MCs (Appendix A). The small non-significant changes are inverse-related to the glucose concentrations indicating a potential link to glucose consumption in the system.

### 3.3. Hif1α and Albumin Protein Expressions across the Oxygen Gradient

Hif1α and albumin protein expressions were investigated by immunohistochemistry staining with red and green fluorescent antibodies, respectively (Figure 3A,B).

Fluorescence image analysis showed that in chamber1, 29.53% ± 4.46% of Hif1α was localized in the cytoplasm and 70.47% ± 4.46% was localized in the nucleus. In comparison, the relative fluorescence intensity of Hif1α in the cytoplasm in chamber8 was 15.98% ± 4.26% and 84.02% ± 4.26% in the nucleus. Thus, the gradient loss of oxygen concentration induced a significant increase in nuclear translocation of Hif1α as indicated by the differences seen from chamber1 to 8 (*p* = 0.02) (Figure 4A). Moreover, the expression of albumin was decreased, as indicated by a reduction in albumin staining (calculated as relative cellular fluorescence) by 43.26% from the first to the last chamber (*p* = 0.02) (Figure 4B).

The differential expression of Hif1α and albumin within the connected MCs indicate that it is possible to induce an oxygen gradient and partial hypoxic conditions by metabolic consumption of oxygen, as clearly visualized by an increased Hif1α nuclear translocation in HepG2 cells. In parallel to the increasing Hif1α nuclear accumulation, the albumin protein levels are reduced.

### 3.4. HIF1A and ALBUMIN mRNA Expressions

Fluorescent in situ hybridization was completed with ALBUMIN and HIF1A antisense mRNA probes directly labeled with green and red fluorescence, respectively (Figure 5).

Data analysis revealed significantly increased HIF1A mRNA expression in HepG2 cells from 7.25% ± 0.45% in chamber1 to 12.26% ± 2.23% in chamber8 (Figure 6A), which indicates a significant increase in HIF1A gene expression (*p* = 0.02) in association with hypoxic condition from chamber1 to 8. On the other hand, ALBUMIN mRNA expression decreased from 32.78% ± 7.2% in chamber1 to 18.71% ± 3.1% in chamber8 (Figure 6B), which correlates well with IF protein data. Furthermore, a significant difference in ALBUMIN mRNA synthesis between HepG2 cells from chamber1 to 8 (*p* = 0.04) was found. It should be considered that the change of ALBUMIN gene expression is in parallel with its protein expression and also correlates with the upregulated HIF1A mRNA and immediate translocation of Hif1α-subunit from the cytoplasm to the nucleus.

### 3.5. Cells Treated with CoCl_2_ or DFX

HepG2 cells were treated in either four connected MCs (normal media) or one MC with the media containing either DFX (250 µM) (Appendix A) or CoCl_2_ (100 µM) (Appendix A) with the same condition as explained before (50 µL/h) for 72 h. Cells were fixed and stained with albumin and Hif1α antibodies. Images were taken from chamber1 and chamber8 (normal media) and chamber1 (DFX- or CoCl_2_-treated cells). Statistical analysis indicated that there was no significant difference between the albumin expression in HepG2 cells in chamber8 (normal media) and chamber1 (CoCl_2_), however, a significant difference was found between chamber1 (normal media) and both chamber8 (normal media) (*p* < 0.0001) and chamber1 (CoCl_2_, (*p* < 0.0001) (Appendix A). Moreover, staining with Hif1α also showed, that there was no significant difference between nuclear or cytosolic Hif1α expression in chamber8 (normal media) and chamber1 (CoCl_2_). However, there was a significant difference between nuclear and cytosolic Hif1α expression in chamber1 (normal media) and chamber1 (CoCl_2_) (*p* = 0.01) (Appendix A). Taken together this data indicates that our MCs culture system induces hypoxia in the final MC similar to chemically induced hypoxia by DFX or CoCl_2_.

## 4. Discussion

In this study, we established a new, relatively simple in vitro method to investigate gene and protein expression as triggered by a cell-mediated oxygen gradient. The microfluidic design with a defined larger cell cultivation area enables the maintenance of larger cell numbers under more homogenous conditions in one chamber, while also providing clear differences in oxygen concentration in adjacent chambers. Thus, a larger number of cells becomes available to analyze the effect of oxygen depletion. It should be noted that the short length of the connecting tubes (3 cm) and the additional sealing of the connectors is essential to obtain the oxygen gradient in the interconnected MCs, to minimize reoxygenation by diffusion through the material.

Production of albumin is an essential feature of hepatocytes in the liver and is predominantly expressed in normal oxygen environments based on their location in the liver acinus zones [27,28,29,30,31]. On the other hand, Hif1α is the most important protein activated and induced in response to the hypoxic condition. Thus, we selected these genes to demonstrate that the multi-chamber microfluidic oxygen gradient system is suitable to obtain a gradient of hypoxic conditions, therefore mimicking natural oxygen availability. Moreover, oxygen sensors, inserted in the microfluidic flow before and after the MCs showed a ~10% reduction in oxygen level in the media by consumption. Interestingly, oxygen measured by the inlet sensor declined over time and reached a steady state after only 48 h. This could be due to initial media gas content and a slow calibration rate from outside the incubator environment to the 5% CO_2_ area inside the incubator. Indeed, such gas diffusion in MCs has been reported before [32]. A reduction from 17.5% to 15.9% in atmospheric oxygen concentration in the flow-through media was enough to increase HIF1A mRNA expression and increase Hif1α protein translocation from the cytoplasm to the nucleus in a gradient-like manner from chamber1 to chamber8 in HepG2 cancer cells. Similar Hif1α cytoplasmic depletion and nuclear accumulation have been reported in other cell types under hypoxic conditions, e.g., in skeletal muscle cells [33]. We know that this limited decrease is quite different from findings with stationary hepatocyte cultivation in which a reduction to 5% or less in atmospheric oxygen was needed to maintain hepatocyte specific metabolic activity [34] and the similarly low amount of available oxygen in liver acini in vivo [16,35]. An explanation for this could be the very slow diffusion of oxygen in the medium and a limited perturbation of the medium in the microfluidic chip system. The oxygen sensors in our microfluidic setup show the reduction in available oxygen in the medium at the inlet and the outlet after passing through the system and reflect the metabolic consumption of oxygen by the cells. The successful creation of a hypoxia gradient in the connected chambers induces different metabolic conditions that are independently visualized by the nuclear translocation of Hif1α, which is comparable to the translocation found upon treatment with DFX and CoCl_2_.

Looking at glucose consumption and ammonia production, it cannot be confirmed that these important media factors contribute to albumin synthesis. We could not find significant differences in the output of ammonia measured in the collected media after single or multiple MCs (Appendix A). Glucose consumption increased depending on the number of connected MCs leaving it unclear whether glucose consumption is a main factor for albumin production. Therefore, HepG2 cells were treated either with CoCl_2_ (100 µM) or DFX (250 µM) in chamber1. Treatment with these hypoxia-inducing chemicals showed an albumin synthesis reduction in comparison to non-treated cells in chamber1 (normal media; Appendix A). The input medium in chamber1 contains equal amounts of glucose for treated and non-treated cells, however, albumin production is reduced with hypoxia mimetic chemicals and inversely correlated with Hif1α translocated from cytoplasm to the nucleus (Appendix A). Taken together, our results support that hypoxia is the main reason for the albumin reduction in the system.

Another important finding of our study is that oxygen depletion seems to be sufficient to reduce albumin expression at the protein and mRNA levels, resembling the albumin distribution found in albumin synthesis in liver zonation [29]. Reduced albumin synthesis due to differences in nutrient and oxygen supply in a microfluidic organ-on-a-chip liver model established with a combination of different liver cell types was reported before [31]. However, such a complex organ model makes it difficult to analyze metabolic adaptation in a larger, defined cell population. With our method reported here, we make it possible to generate compartments with a larger number of cells, with very similar growth conditions and oxygen access that can be used for various downstream analytical methods such as mass spectrometry proteomic or metabolomic profiling, requiring larger cell numbers.

## 5. Conclusions

In this study, we present a simple and versatile in vitro method for the generation of consumption-mediated oxygen gradients to simulate zonation found in liver acinus. Analysis of Hif1α and albumin expression in our model suggests that the zonation of albumin expression in the liver is inversely correlated with Hif1α activity and directly related to the different oxygen levels in the liver cell environment as found along the liver acinus.

## Figures and Tables

**Figure 1 cells-11-03734-f001:**
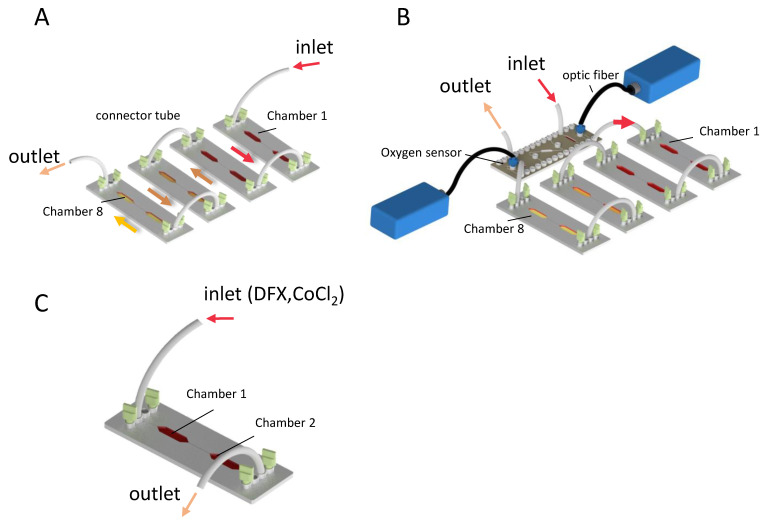
**Serial connection of MCs and oxygen measurement.** Schematic drawing of MCs connected by tubes with oxygen-sealing fitted adapters to obtain eight interconnected microfluidic tissue culture chambers (**A**). Oxygen sensors (Presens OXY-1 ST Fiber optic trace oxygen meter connected to O_2_ SensorPlug and placed in a sensor MC) were inserted before the first inlet and after the final outlet of the connected MCs; (**B**). HepG2 cells were seeded in one MC and treated with 250 µM DFX or 100 µM CoCl_2_ (**C**).

**Figure 2 cells-11-03734-f002:**
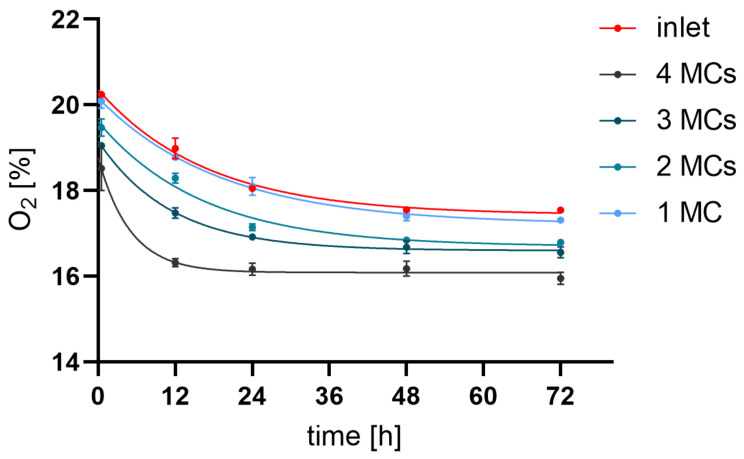
**Oxygen levels in the medium before and after passage through 4 serially connected MCs**. Oxygen percentage measured in the media before chamber1 (inlet) and after chamber2 (1 MC), chamber4 (2 MC), chamber6 (3 MC), and chamber8 (4 MC, outlet) of connected MCs during 72 h of cultivation with constant media flow of 50 µL/h (error bars are not visible for some of the points due to a small standard deviation).

**Figure 3 cells-11-03734-f003:**
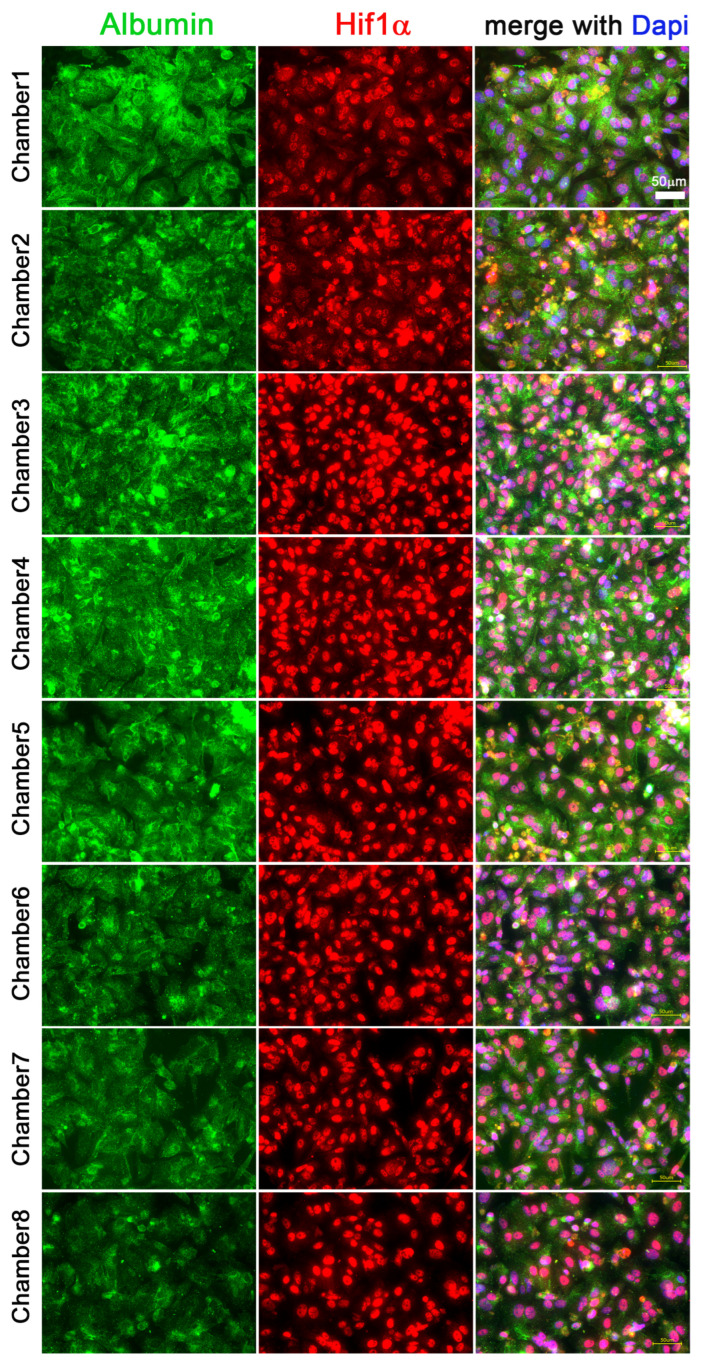
**Albumin and Hif1α immunofluorescence co-staining.** IF staining of HepG2 cells was completed in all eight chambers of connected MCs after 72 h of cultivation with a media flow of 50 µL/h. Albumin is visualized in green and Hif1α is displayed in red. Scale bar 50 µm (**A**). Higher magnification of images for comparison of albumin (in green) and Hif1α (in red) expressed in HepG2 cells in chamber1 and 8 (as in the experiment above). Scale bar 20 µm (**B**).

**Figure 4 cells-11-03734-f004:**
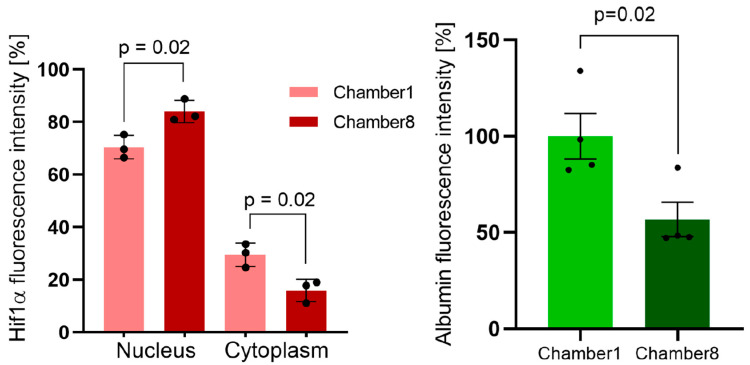
**Quantification of albumin and Hif1α immunofluorescence staining and nuclear translocation.** Albumin and Hif1α fluorescence in HepG2 cells in the connected MCs were quantified in immunofluorescence images from three independent experiments using ImageJ. Graphs were generated using GraphPad Prism. Staining intensities of Hif1α in the nucleus and cytoplasm show increased translocation of Hif1α from cytoplasm to the nucleus when comparing chamber1 and chamber8 with oxygen depletion (*p* = 0.02). Hif1α fluorescence intensity is normalized to 100% total cellular fluorescence (nucleus and cytoplasm) (**A**). The percentage fluorescence intensity of albumin in HepG2 cells in chamber1 and chamber8 shows downregulation of albumin expression from chamber1 to 8 (*p* = 0.02) (**B**).

**Figure 5 cells-11-03734-f005:**
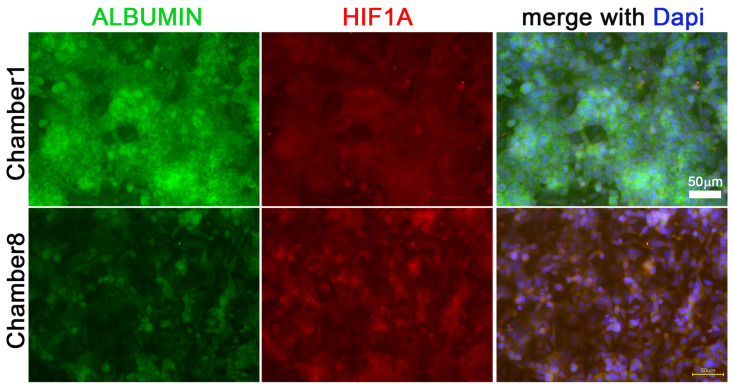
**Fluorescence in situ hybridization (FISH) for ALBUMIN and HIF1A mRNA**. ALBUMIN and HIF1A mRNA expressed in HepG2 cells in chamber1 and chamber8 of connected MCs. Scale bar 50 µm.

**Figure 6 cells-11-03734-f006:**
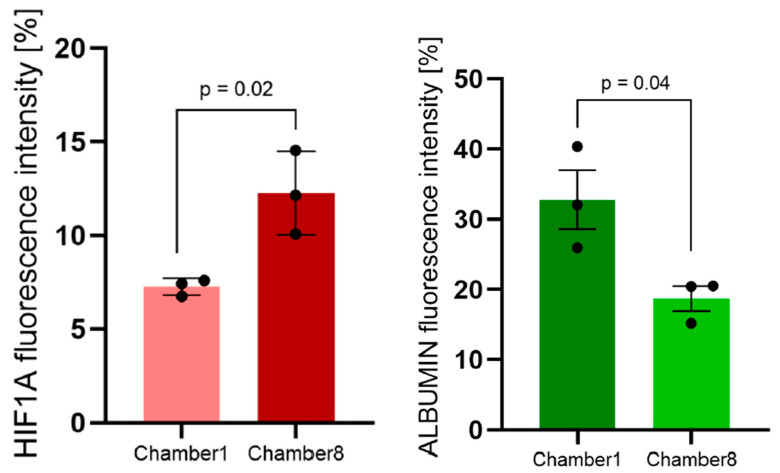
**Quantification of ALBUMIN and HIF1A mRNA fluorescence signals of FISH images.** Relative fluorescence intensity of HIF1A (**A**) and ALBUMIN (**B**) in situ hybridization signals in chamber1 and chamber8 along the oxygen gradient in the connected MCs. The levels of HIF1A mRNA increase from chamber1 to 8 (*p* = 0.02) (**A**); the levels of ALBUMIN mRNA decrease from chamber1 to 8 (*p* = 0.04) (**B**).

**Table 1 cells-11-03734-t001:** List of primary and secondary antibodies used for IF in this study.

Antibodies	Species	Company	Lot	Ratio
Anti-Albumin	Mouse	R&D system, Minneapolis, MN, USA	MAB1455	1:250
Anti-Hif1α	Rabbit (polyclonal)	Atlas antibodies, Bromma, Sweden	HPA001275	1:250
2nd Anti-rabbit _594 nm_	Goat	Sigma-Aldrich, St. Louis, MA, USA	SAB4600107	1:250
2nd Anti-mouse _488 nm_	Goat	Cell signaling, Danvers, MA, USA	4408 s	1:250

**Table 2 cells-11-03734-t002:** List of primers that were used for FISH in this study based on the method introduced before by Ghafoory et al. in 2012 [22].

Gene	NCBI _Reference Sequence_		Primer Sequence
Human ALBUMIN	NM_000477.7	Forward	CAGTGAATTGATTTAGGTGACACTATAGAAGTG**CTGCTGACTTGCCTTCATTAGCTGC**
		Reverse	CAGTGAATTGTAATACGACTCACTATAGGGAGA**CCTGTTCACCAAGGATTCTGTGCAG**
Human HIF1A	NM_001530.4	Forward	CAGTGAATTGATTTAGGTGACACTATAGAAGTG**CATGGAAGGTATTGCACTGCACAGG**
		Reverse	CAGTGAATTGTAATACGACTCACTATAGGGAGA**CAGCACTACTTCGAAGTGGCTTTGG**

**Table 3 cells-11-03734-t003:** Oxygen percentage for inlet and outlet using 1, 2, 3, and 4 MCs at different time points.

Time	Inlet	Outlet			
	All Conditions	1MC	2MCs	3MCs	4MCs
30 min	20.25 ± 0.06	20.09 ± 0.29	19.46 ± 0.34	19.05 ± 0.09	18.52 ± 1.26
12 h	18.99 ± 0.42	18.77 ± 0.08	18.29 ± 0.20	17.48 ± 0.25	16.31 ± 0.23
24 h	18.05 ± 0.10	18.10 ± 0.36	17.15 ± 0.14	16.92 ± 0.16	16.16 ± 0.35
48 h	17.55 ± 0.09	17.40 ± 0.19	16.84 ± 0.07	16.68 ± 0.30	16.18 ± 0.43
72 h	17.55 ± 0.05	17.31 ± 0.12	16.79 ± 0.09	16.56 ± 0.27	15.95 ± 0.36

## Data Availability

Appendix A.

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
