# Peer review of "Oxygen Gradient Induced in Microfluidic Chips Can Be Used as a Model for Liver Zonation"

_cells, 2022, doi:10.3390/cells11233734_

Round 1

Reviewer 1 Report

In this manuscript, the authors present a new, simple approach to generate an oxygen gradient in microfluidic cell culture. Using commercial microfluidic chips attached in serial, a slight oxygen depravation is seen between the first culture chamber and the eighth one. Using hepatocytes, the authors claimed that this oxygen gradient induces a more important localization of the HIFA protein inside the nucleus and that the albumin production is reduced.

While this gradient generation method is definitively of great interest, several main points lead me to recommend a major revision/rejection of the manuscript :

-         First, such serial connection of microfluidic system do not only change the oxygen level in the medium, but also the quantity of glucose, serum factors, waste products, etc etc. Therefore, the biological effect observed (albumin production and localization of HF1A) could not be linked only to an oxygen gradient. Two control experiments could be run : one with different oxygen tension in the incubator, or re-using the medium obtained at the outlet of the 8 chambers in a second experiment (and so subjecting the 1st chamber to a normal level of oxygen but lower amount of glucose and different mix of factors).

-         Better caracterization of the oxygen gradient would be necessary. In particular, what is the oxygen decrease for a chain of 1MC, 2MC, 3MC, 4MC ? Is there any contribution of the tubing length ?

-         In the introduction, a description of the liver geometry and that hepatocyte zonation is necessary, for example with an illustration, to understand the link between oxygen gradient and the in-vivo situation.

-         As the manuscript deals with oxygen control in microfluidic system, a state of the art of the solutions already proposed seems necessary.

-         Description of the microfluidic chips used has to be completed : material (O2 permeability ?), dimensions of the chambers, material of the tubings (O2 permeability ?).

Minor points :

-         In the introduction, the retention of the CDH1 mRNA is not clear and do not bring anything to the manuscript, except an extra self-citation for the authors.

-         Better citations could be found for the first concept of culturing cells in microfluidic systems (l.69)

-         What are the inlets/outlets 1 and 4 ? (l.88)

-         A short description of the principle of the O2 sensors would be useful !

Author Response

We thank the reviewers for their careful consideration, comments and time. Please find our point-by-point response in the word document.

Reviewer 2 Report

The authors have proposed an experimental set up to create oxygen gradients for in vitro cell culture. To this aim, they have serially connected 4 commercially available microfluidic devices and culture the cells in their chamber under continuous flow rate for 3 days. They have also correlated the expressions of HIF1α and albumin to the generated oxygen gradient. The proposed study suffers from the following critical issues:

1) The study claims that the model is suitable for modeling the liver zonation. However, the liver zonation consists of 3 metabolic zones periportal Zone 1, intermediate perivenous Zone 2, perivenous Zone 3 (Ref. : https://doi.org/10.7554/eLife.46206 ). To properly simulate the liver zonation, the oxygen level should be properly controlled and fixed. That is Zone 1 should be supplied with highly oxygenated blood (pO2 ≈ 60–70 mmHg ≈ 10–12%) and Zone 3 needs to receive depleted blood (pO2 ≈ 25–35 mmHg ≈ 3–5%) (Ref. https://doi.org/10.1007/s00204-013-1078-5). However, in this study, the oxygen level cannot be controlled and it is always much higher than those of the above-mentioned liver zones (as shown in Figure 2).

2) It is unclear how this study controls the oxygen level. There are no engineering considerations for this device. For instance, why the flow rate is chosen at 50 ul/h and why 4 devices have been chosen? Also, how the material of the microfluidic device affect these results (e.g., oxygen permeable ones vs impermeable one). The oxygen concentration should be quantified using the well-established methods (see for example, the following papers: https://doi.org/10.1177/1535370217703978; https://doi.org/10.1007/s10544-022-00615-1 and https://doi.org/10.3390/mi9030094)

3) According to Figure 2, the max value of oxygen at the inlet is 17.5% at the inlet and 15.9% at the outlet after 73 hours. How this small difference of 1.6% can affect the HIF1α and albumin expression levels as shown in Figure 4. This difference can be well within the measurement error of the oxygen sensors used at the inlet and outlet.

Minor issues:

1) Please define “MC” throughout the manuscript.

2) Lines 86-89: Does the device has 3 inlets and 3 outlets or 4 inlets and 4 outlets (as the inlets and outlets 1 and 4 have been blocked)?

3) Line 107: What is microfluidic chip number “1090”?

4) What is the concentration of the cells in terms of cells per mL?

5) Although the commercial microfluidic chips have been used, their dimensions and materials should be mentioned in the study.

6) It would be more meaningful, if the actual image of the interconnected devices also be added in Figure 1. 

Author Response

(The authors gave the same response as above.)

Round 2

Reviewer 1 Report

The authors correctly took into account the previous comments.

I still see one point which has to be checked before publication: the material used for the tubings is silicone, and it is known that silicone tubings are incredibly permeable to gas (contrary to polystyrene MCs). It's quite possible that the measurements of O2 under-estimate the real O2 decreasing produced by the cell consumption due to a large increase of O2 concentration between the MCs due to the tubings. Would it be possible to flow medium with a lower O2 concentration into a MC chain without the cells, and check if the O2 concentration changed? This would give the tubings (and MC) contribution to the oxygen level change during the experiments.

Outside this important point, the manuscript would be fine for publication.

Author Response

We like to thank the reviewer for this important comment and the suggestion of a new experiment to demonstrate reoxygenation and analyze the reoxygenation rate in our setup. We discussed the options for doing this experiment and are sure that we could do a suitable experiment in the future. Because the experiment will require a more complex setup, it is not feasible to do these measurements in a time frame for revision. We however like to take up this suggestion to address this point in a new project.

We agree with the reviewer, that the change in O2 concentration observed in our setup, underestimates the consumption by the cells in the system because of reoxygenation, due to the oxygen permeability of the material used. We know that this is an important point because it plays a major role when optimizing the setup for the connections. While the use of additional silicone sealing around the connectors was the most obvious improvement, we also know that the length and geometry of the tubing are important. We, therefore, added additional information in materials and methods to describe the tubing used and the connections of the MCs in more detail.

In addition, we also added a remark on this point in the discussion, to emphasize this point and highlight the importance of the short and sealed connections between the MCs.

Reviewer 2 Report

The authors have carefully replied all my comments. The revised manuscript can be accepted now.

Author Response

We thank the reviewer for accepting our revisions that we prepared based on his initial comments.